# Incidence Rates for Invasive *Streptococcus pneumoniae* and *Haemophilus influenzae* Infections in US Military Pediatric Dependents Before and During COVID-19

**DOI:** 10.3390/vaccines13030225

**Published:** 2025-02-24

**Authors:** Matthew D. Penfold, Sarah Prabhakar, Michael Rajnik, Apryl Susi, Monisha F. Malek, Cade M. Nylund, Elizabeth Hisle-Gorman, Matthew D. Eberly

**Affiliations:** 1Department of Pediatrics, Uniformed Services University of the Health Sciences, Bethesda, MD 20814, USA; sarah.prabhakar.ctr@usuhs.edu (S.P.); apryl.susi.ctr@usuhs.edu (A.S.); cade.nylund@usuhs.edu (C.M.N.); elizabeth.hisle-gorman@usuhs.edu (E.H.-G.); 2The Henry M. Jackson Foundation for the Advancement of Military Medicine, Inc., Bethesda, MD 20817, USA; 3Pediatric Infectious Diseases, Brooke Army Medical Center, Joint Base San Antonio, Fort Sam Houston, San Antonio, TX 78234, USA; monisha.malek@comcast.net (M.F.M.); matthew.eberly@usuhs.edu (M.D.E.)

**Keywords:** *Streptococcus pneumoniae*, *Haemophilus influenzae*, pediatrics, infectious diseases, hospitalization, COVID-19, pandemic

## Abstract

**Background/Objectives**: Invasive *Streptococcus pneumoniae* disease (IPD) and invasive *Haemophilus influenzae* (IHI) infections cause disease in pediatric patients. The COVID-19 pandemic brought about a change in the rates of common viral illnesses that can lead to superimposed bacterial infections. **Methods**: A repeated monthly cross-sectional study was performed using inpatient data from the Military Health System Data Repository (MDR) to observe differences in IPD and IHI hospitalization rates before and during the COVID-19 pandemic starting in March 2018 and continuing to February 2023. Our study included a cohort of 1.27 million children under the age of 5 years old. **Results**: A total of 200 unique cases of IPD and 171 unique cases of IHI were identified. In Year 1 of the pandemic, the hospitalization rates for IHI and IPD decreased. In Year 2, IPD returned to the pre-pandemic baseline, and IHI remained below the baseline. In Year 3, IPD increased above the baseline, and IHI returned to the baseline. **Conclusions**: These data support the notion that the interventions implemented to reduce the spread of COVID-19, such as hand hygiene and social distancing, likely led to a reduction in the incidence of invasive disease. The subsequent relaxation of these mitigation strategies likely led to a resurgence of IHI and an increase in IPD in our population.

## 1. Introduction

Invasive *Streptococcus pneumoniae* disease (IPD) and invasive *Haemophilus influenzae* (IHI) infections cause significant disease in pediatric patients around the world, resulting in bacterial meningitis, bacteremia, and pneumonia in this vulnerable population [1,2]. These pathogens are common nasopharyngeal colonizing bacteria that often give rise to invasive disease following common respiratory viral infections, including respiratory syncytial virus and influenza [3,4]. Since the United States (US) introduced conjugate vaccines against *Streptococcus pneumoniae* and *Haemophilus influenzae* type b (Hib) in 2000 and 1985, respectively, there have been substantial decreases in IPD and IHI [5,6,7]. For example, in a study from 2001, the pre-vaccine IPD rate in US children less than 60 months old was approximately 75 cases per 100,000, and in a 1987 article, the US pre-vaccine IHI rate in children less than 60 months old was 67–131 cases per 100,000 children [8,9]. Data available through the Centers for Disease Control and Prevention’s (CDC) Active Bacterial Core surveillance program demonstrates that the IPD and IHI incidence rates in 2019 for children less than 5 years of age were 9.2 per 100,000 and 2.2 per 100,000 children, respectively [10]. The US, in contrast to many developing nations, has lower monthly incidence rates of IPD and IHI due to widespread vaccination efforts [11].

After COVID-19 was declared a pandemic by the World Health Organization (WHO) in March 2020, many countries implemented non-pharmacological interventions (NPI) such as stay-at-home orders, meticulous hand hygiene, mandatory masking in public, and social distancing to prevent the spread of SARS-CoV-2. The stay-at-home and social distancing orders had unintended negative consequences that reduced access to outpatient clinics, leading to fewer preventative visits and therefore fewer routine childhood vaccinations being given on time [12]. On the other hand, a positive consequence of NPIs was a significant decrease in global cases of IPD and IHI, as *S. pneumoniae* and *H. influenzae* are primarily transmitted via droplets from infected and colonized individuals [7,13,14]. This decline occurred despite the decrease in childhood vaccinations as NPIs led to fewer respiratory illnesses and subsequently fewer secondary bacterial infections at the beginning of the pandemic [14]. Following the relaxation and decreased stringency of these strategies around the world, IPD and IHI have re-surfaced [15,16,17,18]. Currently, there is a paucity of data from the United States addressing the impact of COVID-19 on IPD and IHI hospitalization rates. Our study intended to evaluate this topic. We hypothesized that pediatric patients enrolled in the Military Health System (MHS) experienced a decrease in access to immunizations that resulted in changes from the pre-pandemic baselines for IPD and IHI. We also evaluated demographic risk factors within our population for IPD and IHI.

## 2. Materials and Methods

### 2.1. Study Design

Inpatient data from the MHS was used to perform a repeated monthly cross-sectional study evaluating differences in inpatient *Streptococcus pneumoniae*, *Haemophilus influenzae*, and Group B *Streptococcus* (GBS) rates and demographic risk factors before and during the COVID-19 pandemic. We sought to compare vaccine-preventable diseases to non-seasonal and non-vaccine-preventable diseases over the same time period by investigating the rates of GBS infections in 0–5-month-old children. In contrast to seasonal respiratory viruses that are frequently followed by peaks of secondary bacterial infections, invasive GBS infections in neonates occur uniformly throughout the year and represent a non-respiratory pathogen. There are no FDA-approved vaccinations against GBS, and NPIs do not alter GBS transmission; as such, GBS provided baseline control of a common invasive infection during the COVID-19 pandemic.

### 2.2. Study Procedure

Data were collected from March 2018 to February 2023 with the first two years (March 2018 to February 2020) considered pre-pandemic and the following three years (March 2020 to February 2023) divided into Year 1, Year 2, and Year 3. Inclusion criteria were children 0–59 months of age within the MHS diagnosed with IPD and IHI during the study period. For the full study population, demographics were recorded at the age of the first encounter in the MHS, whereas demographics for the unique cases were recorded at the time of an IPD or IHI diagnosis. Age was broken into 6 groups: 0–5 months old, 6–11 months old, 12–23 months old, 24–35 months old, 36–47 months old, and 48–59 months old. We additionally included demographic data based on the child’s parental military rank as a proxy for socioeconomic status within our population. In order to evaluate potential regional differences in the delivery of medical care and disease incidence, we utilized the four geographical regions in which the MHS’s primary insurance carrier, TRICARE, is broken into: North, South, West, and Overseas. Statistical analysis was performed utilizing SAS v9.4, Cary, NC, USA.

### 2.3. Data Analysis

Table 1 contains the International Classification of Diseases, Tenth Version (ICD-10) codes that were used to identify IPD, IHI, and GBS diagnoses. Monthly case rates were calculated by taking the number of unique diagnoses of infection seen in a given month divided by the number of children eligible/enrolled for care in that same month.

Rate ratios (RR) with 95% confidence intervals (CI) of IPD, IHI, and GBS were calculated using unadjusted and adjusted Poisson regression with time period at diagnosis and several demographic variables, including sex, parent military rank, age categories, and region.

This study was conducted according to the guidelines of the Declaration of Helsinki; the use of the data for this study was approved by the Uniformed Services University of the Health Sciences Institutional Review Board (USUHS:2020-065—2 October 2021). An HIPAA waiver and waiver of informed consent were obtained because the study utilized de-identified, pre-existing data from the Military Health System database.

## 3. Results

### 3.1. Invasive Pneumococcal Disease

Across the cohort of 1.27 million MHS beneficiaries less than 5 years of age, there were 200 unique cases of IPD in children (Table 2). During the first year of COVID-19, there was a significant decrease in IPD rates compared to pre-pandemic levels with an adjusted RR of 0.31 (95% CI: 0.17, 0.55). During the second year of COVID-19, the rates of IPD returned to the pre-pandemic levels with an adjusted RR of 1.11 (95% CI: 0.78, 1.59). During the third year of the pandemic, the IPD rates increased above the pre-pandemic baseline with an adjusted RR of 1.77 (95% CI: 1.29, 2.42) (Figure 1a and Figure 2a). The monthly cases and rates per 100,000 are depicted in Figure 1a. The forest plot in Figure 2a provides a visual representation of Table 3a that highlights the relative significance of the rate ratios for the variables we evaluated. It demonstrates that the relative increased risk at younger ages for IPD, as well as the significant decrease in IPD cases in the first year, returned to baseline in year two and then significantly increased in the third year. For age, significant adjusted risk ratios were found in patients 0–5, 6–11, 12–23, and 24–35 months old, but not in those 36–47 months old compared to those 48–59 months old. The female sex was associated with a decreased risk of IPD with an adjusted RR of 0.74 (95% CI: 0.57, 0.99). Parent rank, as a marker of socioeconomic status, did not affect the risk of IPD. Patients living within the TRICARE West Region were found to have a higher rate of IPD when compared to the South with an adjusted RR of 1.44 (95% CI: 1.03, 2.02) (Table 3). The most common diagnostic codes were J13 (pneumonia; 42.2%), B95.3 (*Streptococcus pneumoniae* as the cause of diseases classified elsewhere; 33.5%), and A40.3 (sepsis; 16.2%) (Table 4).

### 3.2. Invasive Haemophilus influenzae

Across the same cohort as IPD, there were 171 unique cases of IHI disease in children younger than 5 years old. During the first year of COVID-19, there was a significant 81% decrease in cases of IHI compared to the pre-pandemic level and an adjusted RR of 0.19 (95% CI: 0.10, 0.39) (Table 3, Figure 2). Even during the second year of COVID-19, the rates of IHI were significantly lower by 58% when compared to the pre-pandemic era with adjusted RR of 0.42 (95% CI: 0.26, 0.69). During the third year of the pandemic, the IHI hospitalization rate returned to the pre-pandemic baseline with an adjusted RR of 1.19 (95% CI: 0.85, 1.67). The monthly cases and rates per 100,000 are depicted in Figure 1a. The forest plot in Figure 2b provides a visual representation of Table 3b that highlights the relative significance of the rate ratios for the variables we evaluated. In particular, it shows the higher risk for younger children and the return of IHI infections to the pre-pandemic baseline. For age, significant adjusted rate ratios were found in patients aged 0–5, 6–11, and 12–23 months, but not in those aged 24–35 and 36–47 months compared to those who were 48–59 months of age. Child sex, military sponsor’s rank in the military, and region of enrollment did not affect the rate of IHI (Table 2 and Table 3). The most common diagnostic codes utilized were B96.3 (*Haemophilus influenzae* as the cause of diseases classified elsewhere; 49.9%) and J14 (pneumonia; 37%) (Table 4).

### 3.3. Invasive GBS

There were 165 unique cases of neonatal GBS reported during the study period (Table 2b). The monthly cases and rates per 100,000 infants less than six months are depicted in Figure 1b. There is a non-statistically significant trend towards increased cases of GBS in both Years 1 and 2 of the COVID-19 pandemic, with the highest number of cases being found in Year 2. Interestingly, there was a significant statistical difference during Year 3, showing a decrease in GBS-related hospitalizations when compared to the pre-pandemic baseline with an adjusted RR of 0.62 (95% CI: 0.38, 0.99), which is depicted in the forest plot in Figure 2c.

## 4. Discussion

This cross-sectional study assessed the monthly rates of IPD, IHI, and GBS two years before and three years after the start of the COVID-19 pandemic in pediatric patients enrolled in the MHS. We observed a statistically significant decrease in both IPD and IHI in the first year of the COVID-19 pandemic. This finding is consistent with observational studies across the globe during the same time period which showed fewer invasive bacterial infections in children from IPD and IHI [7,13,19,20,21,22]. This likely reflects the successful impact of NPIs at reducing the spread of all respiratory illnesses, including SARS-CoV-2, and subsequent reduction in hospitalizations for common respiratory pathogens and secondary bacterial infections at the beginning of the pandemic [23,24].

In the second year following the start of the COVID-19 pandemic, IPD hospitalization rates returned to our population’s pre-pandemic baseline, while IHI rates remained low. In the third year, IPD hospitalization rates were above the pre-pandemic baseline, and IHI rates returned to the baseline. This is consistent with global reports that indicate that the relaxation of NPIs led to the return of IPD and IHI [15,16,17,18]. One observational study from Germany showed a 90% increase in IHI during the first quarter of 2022 when compared to pre-pandemic seasonal trends [25]. Data from the CDC’s Active Bacterial Core surveillance appears to show a return to baseline rates in the US for IPD and IHI through 2022 [10]. As far as we are aware, this would make our study the first to show a rise in IPD rates in the United States following the onset of the COVID-19 pandemic.

It has been previously reported that during the COVID-19 pandemic, there were missed early childhood immunizations in both military populations and the civilian US population [26,27]. This likely worsened the disparity in early childhood immunizations, which already existed in the US in 2019. One study found that one in six children did not complete at least one dose of a multidose series in the year preceding the COVID-19 pandemic [28]. In the setting of this vaccination gap, our study showed an increase in IPD rates during the third year but did not show a significant rise above pre-pandemic IHI rates. This is likely multifactorial. Children who missed *Haemophilus influenzae* type b vaccines at a young age may have become less susceptible to invasive disease simply by being older as the risk of severe infection declines with age. This is supported by our data and many other studies indicating that the highest risk for IPD and IHI is <2 years of age [29,30]. It is also possible that our cohort’s catch-up immunizations were completed but have yet to be studied and documented. However, in terms of pneumococcal vaccination, it is certainly plausible that catch-up immunizations were not completed given the rise in hospitalization rates in our data.

There is also likely a factor of herd immunity from children that continued to receive their childhood vaccinations on time that may have protected more children from developing IPD or IHI. When the pneumococcal conjugated vaccine (PCV) series was first released to US children in 2000, studies showed that adults and children who were not immunized also benefited from the herd immunity against *S. pneumoniae* [6,31]. The benefit of herd immunity has also been born out for countries consistently using Hib vaccines [32]. If, in fact, children enrolled in the MHS did not catch up on their immunizations, further monitoring for IPD and IHI cases will be an important area for future research. For now, our study supports the notion that the relaxation of NPIs and return of common viral respiratory infections, along with likely other difficult-to-study factors, fueled the return of secondary invasive bacterial infections caused by IPD and IHI in children enrolled in the MHS. We postulate that the upsurge of respiratory syncytial virus cases during the 2022–2023 season factored into the return of invasive bacterial infections in our population as both IPD and IHI rates drastically increased during this time period.

Overall, invasive GBS disease in neonates did not significantly change during the COVID-19 pandemic as these infections remained stable through the first three years of the pandemic when compared to pre-pandemic years. The third year shows a slight reduction in GBS cases when compared to the baseline but is unlikely to be related to any NPIs. This suggests that GBS screening and labor and delivery protocols were not significantly disrupted during the pandemic, that laboratories were able to appropriately process specimens, and that medical providers were able to continue to accurately code laboratory test results in the MHS during the pandemic. Additionally, this provided control for invasive infections in children, which are not impacted by non-pharmacological interventions or vaccinations, but could be affected by interruptions in access to health care.

Prior to the introduction of the pneumococcal conjugate vaccine (PCV), children younger than 5 years old were particularly vulnerable and saw significantly higher rates of invasive infections when compared to adults. The PCV series has undergone multiple iterations, with each updated vaccine including more serotypes that are known to commonly cause invasive disease. Starting in 2000, PCV has targeted 7, 10, 13, and 15 highly virulent strains of *S. pneumoniae* out of over 100 identified pneumococcal serotypes [33]. PCV-20 was approved by the Advisory Committee on Immunization Practices in 2023 and would not have significantly impacted the data in this study [34]. The *H. influenzae* vaccine only protects against encapsulated Hib, but not types a, c, d, e, f, or the non-typeable *H. influenzae* strains. Prior to the vaccine, Hib was responsible for more than 95% of all *H. influenzae* infections [35]. Since the Hib vaccine’s global implementation, there has been a dramatic decrease in invasive infections around the world caused by Hib with a relative rise in invasive disease caused by non-typeable *H. influenzae* and *H. influenzae* type a [36,37]. Therefore, the impact of delayed vaccination may have less impact on overall IHI since many of the cases are not vaccine preventable. GBS infections are caused by the Gram-positive coccus *Streptococcus agalactiae* and lead to invasive disease in neonates, including bacteremia, meningitis, and pneumonia. Unlike *S. pneumoniae* and *H. influenzae*, there is no vaccine that is currently available for the prevention of invasive GBS disease, and even peripartum antibiotics are only shown to reduce the risk of early-onset disease and not late-onset disease [38]. GBS is also not impacted by seasonality and NPIs in the same way that IPD and IHI are, and it is not known to cause super-imposed bacterial disease following a preceding viral illness.

Asymptomatic colonization with the potential pathogen is typically expected to occur first before causing IPD or IHI. Young children can be colonized with *S. pneumoniae* and *H. influenzae* within their naso- and oro-pharynx and are commonly co-carried [1,2,39]. Additional studies during the pandemic have shown that pediatric patients who were positive by real-time polymerase chain reaction (PCR) testing for SARS-CoV-2 were more likely to be colonized with *S. pneumoniae*, theoretically elevating their risk for IPD [40]. A separate study from China found an increase in *H. influenzae* detection from pharyngeal samples of hospitalized children after the relaxation of COVID-19 NPIs and the return of school [41]. Given the findings in those studies, it is possible that children in our population were colonized with *S. pneumoniae* or *H. influenzae* at a higher rate later in the pandemic and therefore increased the number of patients who had the potential to acquire IPD or IHI.

Studies are mixed on whether NPIs and vaccination reduce colonization with *S. pneumoniae* and *H. influenzae*. One study showed higher rates of asymptomatic pharyngeal colonization of *S. pneumoniae* among children who attend daycare and children who were fully immunized against pneumococcus [42]. Other studies showed that vaccination was associated with lower colonization rates of *S. pneumoniae* [42,43]. Of those, Koliou et al. also demonstrated an increased risk of colonization with daycare exposure and having siblings [43]. Still, prior to the pandemic, others showed either no difference in *S. pneumoniae* colonization rates between unvaccinated and fully vaccinated children or that the colonizing serotypes were shifted to non-PCV strains [44,45]. Another study from Belgium indicated that NPIs during the pandemic did not affect pneumococcal carriage in infants but still reduced the incidence of IPD [46]. Finally, another study from the US showed decreased *H. influenzae* colonization during the first 9 months of the pandemic and a shift to *S. pneumoniae* colonization with more antibiotic-resistant serotypes, but there were fewer outpatient visits for acute otitis media and upper respiratory infections [47].

It is reasonable to postulate that during the time of strict adherence to NPIs there was clinically relevant decreased transmission of *S. pneumoniae* and *H. influenzae*, along with decreased transmission of common respiratory viruses, which, as a whole, led to fewer secondary invasive bacterial infections among our population. This is supported by the well-documented drop in invasive bacterial infections globally when NPIs were widely implemented despite the conflicting evidence about the impact of NPIs and vaccinations on bacterial pharyngeal colonization.

As mentioned previously, the population that this study includes is spread across the entirety of the United States and the globe, which means that the intensity and duration of the NPIs that impacted this cohort were asynchronous and unpredictable. In general, NPIs evolved and changed and were driven by vaccine availability, natural immunity, public tolerance of restrictions, and the expectation for children to return to school and adults to work. Specific to the United States, the first year of the pandemic was defined by widespread lockdowns and stay-at-home orders. In the second year of the pandemic, as vaccines became more widespread, NPIs loosened and children returned to school, albeit in modified formats for many. In the third year and to date, nearly all facets of travel, work, school, and social activities have returned to their pre-pandemic forms. The loosening of NPIs and return to pre-pandemic lifestyles with the expected transmission of common respiratory pathogens likely played a role in the steady return of IPD and IHI.

Our study’s primary strength is the size of the cohort. We were able to evaluate diagnostic codes for over 1.27 million MHS beneficiaries under 5 years of age who received care either through military treatment facilities or civilian medical institutions. Another strength is that the military pediatric population represents a diverse cross-section of US children, those who have free access to health care, and those who live across the globe and therefore offers a unique opportunity to study vaccine-preventable diseases, such as IPD and IHI. This study also included several limitations, including its reliance on ICD-10 codes, and therefore provided coding across military and civilian medical facilities. This would have caused cases to be missed if non-pathogen-specific ICD-10 codes were utilized by providers. Specific to GBS, this limited our ability to differentiate between early- and late-onset GBS disease. Additionally, this study used population enrollment data in the health care system for all three pathogens and did not include live birth data to determine the rates of GBS disease. Another limitation of our monthly cross-sectional design was the difficulty in reporting the number of unique individuals in a single table as the population changed on a monthly basis. For this reason, the full study’s population numbers represent the age at which the population entered the MHS, but the ages of unique cases that were updated and represent the age at which a child received an ICD-10 code commensurate with our study criteria. Our study did not include laboratory testing such as culture, serotyping, or molecular testing to confirm the ICD-10 codes. This limited our ability to identify cases whose ICD-10 codes did not include a pathogen, and we could not confirm ICD-10 codes that included a pathogen. In terms of NPIs, given the global distribution and wide variance in local and national ordinances, it is difficult to know whether our cohort experienced more or less restrictive lifestyle changes as a result of the pandemic. Finally, we did not utilize individual patient data and were therefore unable to evaluate the underlying medical conditions, risk factors, and vaccination statuses of the hospitalized patients at the time of their admission. This limited the study’s ability to provide more in-depth analyses beyond the demographics in Table 2 and Table 3.

## 5. Conclusions

There was a significant decrease in both IPD and IHI in the first year of the COVID-19 pandemic. In the second year, IPD rates returned to our population’s pre-pandemic baseline, while IHI rates remained low. In the third year, IPD rates increased above the baseline, and IHI rates returned to the pre-pandemic levels. GBS hospitalization rates in infants remained steady throughout the 5 years of our study with a minimal decrease during Year 3 of the pandemic.

These data support the notion that non-pharmacological interventions and measures meant to reduce the spread of COVID-19, such as the implementation of masks, social distancing, quarantining, and school closures, likely led to subsequent reductions in colonization, respiratory transmissions, and incidences of invasive bacterial disease initially. However, with the relaxation of these mitigation strategies and missed routine childhood immunizations, a resurgence of IHI cases and an increase in IPD cases were observed in our population. As far as we are aware, this is the first report of increased IPD hospitalization rates in a predominantly United States-based pediatric population. Given this resurgence in disease, appropriate and timely childhood immunizations are paramount to protecting this vulnerable population.

## Figures and Tables

**Figure 1 vaccines-13-00225-f001:**
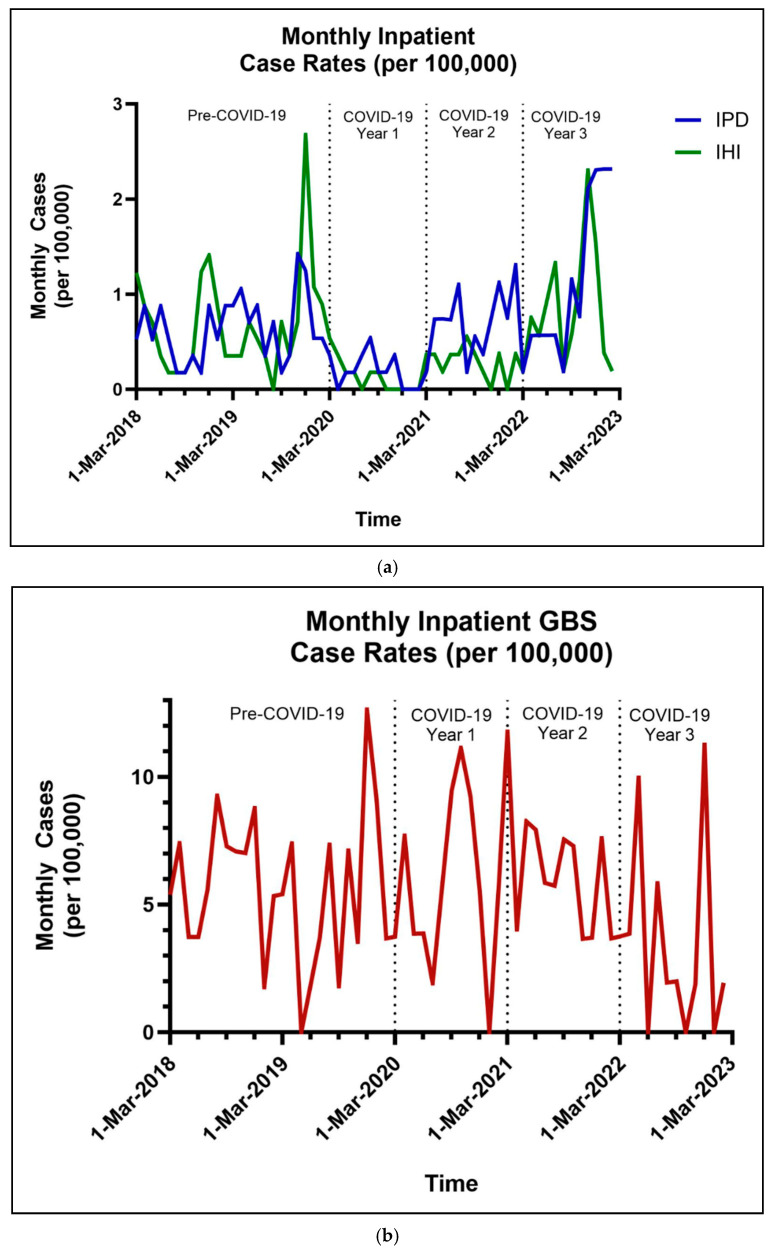
(**a**) Monthly rates of invasive pneumococcal disease and invasive *Haemophilus influenzae*. (**b**) Monthly rates of Group B Streptococcus.

**Figure 2 vaccines-13-00225-f002:**
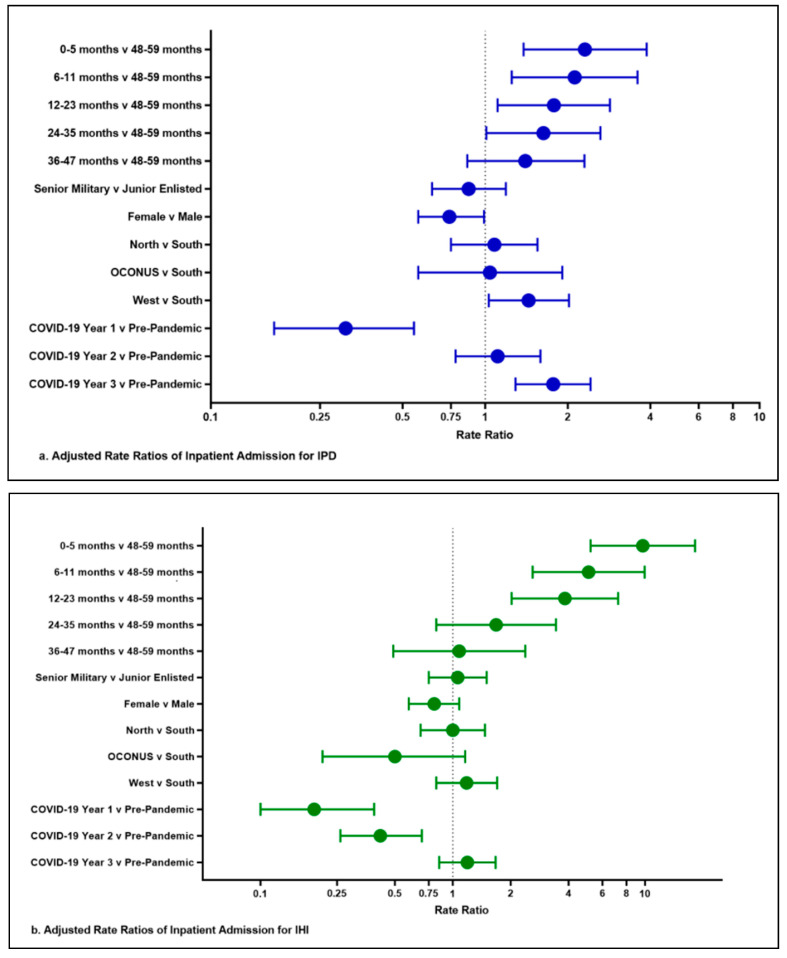
(**a**–**c**) Invasive pneumococcal disease, invasive *Haemophilus influenzae*, and Group B Streptococcus forest plots of adjusted rate ratios for inpatient admissions.

**Table 1 vaccines-13-00225-t001:** ICD-10 codes.

*Streptococcus pneumoniae* codes:
G00.1 (meningitis), A40.3 (sepsis), J13 (pneumonia), B953, M0010, M00111, M00112, M00119, M00121, M00122, M00129, M00131, M00132, M00139, M00141, M00142, M00149, M00151, M00152, M00159, M00161, M00162, M00169, M00171, M00172, M00179, M0018, M0019
*Haemophilus influenzae* codes:
G00.0 (meningitis), A41.3 (sepsis), J14 (pneumonia), A492, B963
*Streptococcus agalactiae* (Group B Streptococcus) codes:
P36.0 (newborn sepsis), P23.3 (congenital pneumonia), A40.1 (newborn sepsis), B95.1 (cause of diseases classified elsewhere [meningitis]), J153 (pneumonia)

**Table 2 vaccines-13-00225-t002:** (a) Demographics for IPD and IHI at time of first encounter. (b) Demographics for GBS at time of first encounter.

**(a)**				
		**Unique Children with IPD (*n* = 200) ***	**Unique Children with IHI (*n* = 171) ***	**Full Study Population (*n* = 1,268,604) ****
**Age (months old)**	0–5 months	31 (15.5%)	56 (32.8%)	604,871 (47.7%)
	6–11 months	26 (13.0%)	29 (17.0%)	80,395 (6.3%)
	12–23 months	43 (21.5%)	42 (24.6%)	152,975 (12.1%)
	24–35 months	40 (20.0%)	20 (11.7%)	148,146 (11.7%)
	36–47 months	34 (17.0%)	12 (7.0%)	143,572 (11.3%)
	48–59 months	26 (13.0%)	12 (7.0%)	138,645 (10.9%)
**Sex**	Male	118 (59.0%)	97 (56.7%)	648,834 (51.2%)
	Female	82 (41.0%)	74 (43.3%)	619,770 (48.9%)
**Sponsor Rank**	Junior Enlisted	50 (25.0%)	40 (23.4%)	369,208 (29.1%)
	Other	150 (75.0%)	131 (76.6%)	899,396 (70.9%)
**Region**	North	59 (29.5%)	53 (31.0%)	392,933 (31.0%)
	South	52 (26.0%)	49 (28.7%)	376,091 (29.7%)
	West	77 (39.0%)	63 (36.8%)	409,512 (32.3%)
	Outside the US	12 (6.0%)	6 (3.5%)	90,068 (7.10%)
**(b)**				
		**Unique Children with GBS (*n* = 165)**		**Full Study Population (*n* = 533,911)**
**Sex**	Male	84 (50.9%)		309,598 (51.2%)
	Female	81 (49.1%)		295,475 (48.8%)
**Sponsor Rank**	Junior Enlisted	48 (29.1%)		184,869 (30.6%)
	Other	117 (70.9%)		420,204 (69.4%)
**Region**	North	37 (22.4%)		182,931 (30.2%)
	South	68 (41.2%)		174,994 (28.9%)
	West	52 (31.5%)		201,933 (33.4%)
	Outside the US	8 (4.9%)		45,215 (7.5%)

***** Age at first IPD or IHI hospitalization. ****** Age at entry into MHS.

**Table 3 vaccines-13-00225-t003:** (a–c) Invasive pneumococcal disease, invasive *Haemophilus influenzae*, and Group B Streptococcus, IHI, and GBS inpatient results—unadjusted and adjusted risk ratios (RRs).

**(a) IPD Unadjusted and Adjusted Risk Ratios (RRs)**			
		**Unadjusted RR (95% CI)**	**Adjusted RR (95% CI)**
**Age (Reference = 48–59 months)**	0–5 months	2.40 (1.43, 4.02)	2.31 (1.38, 3.88)
	6–11 months	2.19 (1.30, 3.70)	2.12 (1.25, 3.59)
	12–23 months	1.81 (1.13, 2.90)	1.78 (1.11,2.85)
	24–35 months	1.65 (1.02, 2.66)	1.63 (1.01, 2.63)
	36–47 months	1.41 (0.86, 2.32)	1.40 (0.86, 2.30)
**Rank (Reference = Junior Enlisted)**	Senior Military	0.82 (0.60, 1.11)	0.87 (0.64, 1.19)
**Sex (Reference = Male)**	Female	0.74 (0.57, 0.97)	0.74 (0.57, 0.99)
**Region (Reference = South)**	North	1.08 (0.75, 1.55)	1.08 (0.75, 1.55)
	Outside the US	1.04 (0.57, 1.89)	1.04 (0.57, 1.91)
	West	1.45 (1.03, 2.03)	1.44 (1.03, 2.02)
**Time Period (Reference = Pre-COVID-19)**	COVID-19 Year 1	0.31 (0.17 0.55)	0.31 (0.17, 0.55)
	COVID-19 Year 2	1.11 (0.78, 1.59)	1.11 (0.78, 1.59)
	COVID-19 Year 3	1.76 (1.29, 2.41)	1.77 (1.29, 2.42)
**(b) Invasive Haemophilus influenzae Unadjusted and Adjusted Risk Ratios (RRs)**			
		**Unadjusted RR (95% CI)**	**Adjusted RR (95% CI)**
**Age (Reference = 48–59 months)**	0–5 months	9.75 (5.22, 18.18)	9.73 (5.21, 18.19)
	6–11 months	5.10 (2.61, 9.96)	5.08 (2.60, 9.94)
	12–23 months	3.82 (2.02, 7.23)	3.83 (2.02, 7.24)
	24–35 months	1.69 (0.82, 3.45)	1.68 (0.82, 3.44)
	36–47 months	1.09 (0.50, 2.38)	1.08 (0.49, 2.38)
**Rank (Reference = Junior Enlisted)**	Senior Military	0.84 (0.60, 1.19)	1.06 (0.75, 1.50)
**Sex (Reference = Male)**	Female	0.80 (0.59, 1.08)	0.80 (0.59, 1.08)
**Region (Reference = South)**	North	1.01 (0.69, 1.49)	1.00 (0.68, 1.47)
	Outside the US	0.52 (0.23, 1.22)	0.50 (0.21, 1.16)
	West	1.22 (0.85, 1.76)	1.18 (0.82, 1.70)
**Time Period (Reference = Pre-COVID-19)**	COVID-19 Year 1	0.19 (0.10, 0.38)	0.19 (0.10, 0.39)
	COVID-19 Year 2	0.42 (0.26, 0.69)	0.42 (0.26, 0.69)
	COVID-19 Year 3	1.21 (0.86, 1.69)	1.19 (0.85, 1.67)
**(c) Group B *Streptococcus* Unadjusted and Adjusted Risk Ratios (RRs)**			
		**Unadjusted RR (95% CI)**	**Adjusted RR (95% CI)**
**Rank (Reference = Junior Enlisted)**	Senior Military	1.04 (0.75, 1.45)	1.06 (0.76, 1.48)
**Sex (Reference = Male)**	Female	1.04 (0.77, 1.40)	1.04 (0.77, 1.40)
**Region (Reference = South)**	North	0.53 (0.36, 0.78)	0.53 (0.36, 0.78)
	Outside the US	0.50 (0.25, 1.00)	0.50 (0.25, 0.99)
	West	0.66 (0.46, 0.94)	0.66 (0.46, 0.94)
**Time Period (Reference = Pre-COVID-19)**	COVID-19 Year 1	1.00 (0.67, 1.49	1.00 (0.67, 1.48)
	COVID-19 Year 2	1.12 (0.77, 1.65)	1.12 (0.76, 1.64)
	COVID-19 Year 3	0.62 (0.39, 1.00)	0.62 (0.38, 0.99)

**Table 4 vaccines-13-00225-t004:** ICD-10 code counts and percentages for invasive pneumococcal disease and invasive *Haemophilus influenzae*.

**Invasive Pneumococcal Disease**	**Count (*n* = 526)**	**Percentage (%)**
J13 (pneumonia)	222	42.21
B95.3 (*Streptococcus pneumoniae* as the cause of diseases classified elsewhere)	176	33.46
A40.3 (sepsis)	85	16.16
G00.1 (meningitis)	39	7.41
M00.112 (pneumococcal arthritis, left shoulder)	2	0.38
M00.151 (pneumococcal arthritis, right hip)	2	0.38
**Invasive *Haemophilus influenzae***	**Count (*n* = 373)**	**Percentage (%)**
B96.3 (*Haemophilus influenzae* as the cause of diseases classified elsewhere)	186	49.86
J14 (pneumonia)	138	37
A41.3 (sepsis)	27	7.24
G00.0 (meningitis)	20	5.36
A49.2 (*Haemophilus influenzae* infection, unspecified site)	2	0.54

## Data Availability

All data were accessed from the Military Health System database, which requires a valid data sharing agreement due to the nature of personal health information and cannot be shared without periodic approval from the US Department of Defense, Defense Health Agency.

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
