# Peer review of "Incidence Rates for Invasive Streptococcus pneumoniae and Haemophilus influenzae Infections in US Military Pediatric Dependents Before and During COVID-19"

_vaccines, 2025, doi:10.3390/vaccines13030225_

Round 1

Reviewer 1 Report

Comments and Suggestions for Authors

Below are my points:

the manuscript provides an insightful examination of pediatric infectious diseases within a unique population during the COVID-19 pandemic. 

While the introduction effectively sets up the study's rationale, it would benefit from a deeper exploration of how the pandemic specifically impacted pediatric healthcare access and community interactions, which are crucial for understanding shifts in disease incidence.

The selection of the age range needs further justification, particularly why children older than this range were excluded, as this could significantly affect the study’s applicability to broader age groups. Additionally, the analysis would be strengthened by accounting for variables such as underlying health conditions or socioeconomic factors beyond parental military rank, which could influence the outcomes.

The manuscript could also benefit from a more detailed year-by-year analysis to clarify why certain trends appear in specific years, discussing any related public health measures or changes within the military community.

The discussion of limitations currently lacks depth.

Specifically, how the demographics of the study population correlate with the findings and the trends depicted in the visual data need to be articulated more cohesively. 

Sincerely.

Author Response

Comment 1: "the manuscript provides an insightful examination of pediatric infectious diseases within a unique population during the COVID-19 pandemic."

Response 1: Thank you for your kind words. This was a great research project for our team to collaborate together on.

Comment 2: "While the introduction effectively sets up the study's rationale, it would benefit from a deeper exploration of how the pandemic specifically impacted pediatric healthcare access and community interactions, which are crucial for understanding shifts in disease incidence."

Response 2: We certainly agree that the pandemic's broad impacts across all sectors of medicine and life affected how disease incidence shifted during that time. In the introduction, we have included a citation of the American Academy of Pediatrics COVID-19 working group that highlights the negative impact the pandemic had on preventative health and immunization visits at physicians' offices.

Comment 3: "The selection of the age range needs further justification, particularly why children older than this range were excluded, as this could significantly affect the study’s applicability to broader age groups. Additionally, the analysis would be strengthened by accounting for variables such as underlying health conditions or socioeconomic factors beyond parental military rank, which could influence the outcomes."

Response 3: We appreciate this very prudent comment - in particular about ways our study's analysis could be strengthened. Unfortunately, there are many limitations to accessing all of the data needed within our large military heath system's data repository. While having more granular patient details would be immensely useful, we are unable to get patient specific data and therefore are unable to further account for underlying medical comorbidities, immunization status, or SES factors. In terms of the age parameters, this study was done as part of an even larger study that looked at all childhood immunizations up through the age at which children enter the public school system in the United States (usually around 5 years of age).  

Comment 4: "The manuscript could also benefit from a more detailed year-by-year analysis to clarify why certain trends appear in specific years, discussing any related public health measures or changes within the military community."

Response 4: We agree that this is an important part of what our study evaluated.  What makes this difficult in our cohort is our military population is spread across the globe, and while the military can dictate what service members are required to do, local, state, and national ordinances would have dictated what the families and children experienced.  The NPIs that would have been enforced would have depended on where a child was living during the study period, which would further be complicated by our population regularly changing duty locations.  To that end, in the discussion section, we added a general description of how public health measures evolved in the United States during the study period, acknowledging that our population may not have experienced these in the same manner or at the same time.

Comment 5: "The discussion of limitations currently lacks depth."

Response 5: Any good study can acknowledge its limitations, and we appreciate the reviewer pointing out that we could provide more context to the limitations we listed. Additional comments were added to that section of the discussion to meet that end.

Comment 6: "Specifically, how the demographics of the study population correlate with the findings and the trends depicted in the visual data need to be articulated more cohesively."

Response 6: Thank you for this insight and certainly this was an oversight on our part as the forest plots were hardly acknowledged at all in the results. We have added a few comments in the results section that hopefully points future readers to the utility of those figures.

Reviewer 2 Report

Comments and Suggestions for Authors

This is a good large-scale study assessing the changes in IPD and IHI rates using GBS infection rates as a reference. Several issues need to be addressed.

1. Line 10: The abbreviation "IPD" doesn't clearly reflect the text behind it. Could you please clarify?

2. Line 15: Add the year periods.

3. In the conclusion, you mentioned "interventions implemented to reduce the spread of COVID-19." Please provide some examples.

4. Line 56: Spell out "MHS"

5. Methods: Comparing IPD and IHI incidence rates to those if GBS wasn't described in the abstract. It seems like this is an essential part of your study since GBS infections are not prevented by vaccines.

6. Line 71: How did you define the first two years? From March 2018 to February 2020? Please clarify in between parentheses.

7. Line 76: Citation of results tables should be limited to the results section. Please remove it from this line.

8. Lines 83-85: Please provide a reference for this sentence.

9. Line 86: I suggest making Table 3 as table 1 and moving it to follow this paragraph.

10. Lines 89-92: Separate this part into a standalone paragraph.

11. Methods: it's better to divide this section into subsections with subheadings.

12. Line 97: rate of care or incidence rate?

13. Table 1: Why is the full study population different between table 1a and table 1b?

14. There's no description in the results for Figure 2.

15. Line 210 and 224: Spell out "RSV" and "PCV"

16. Lines 231-234: This sentence describing H. influenzae should be moved to the introduction.

17. Conclusion: Add a better emphasis on the importance of childhood vaccines at the end of the second paragraph.

Author Response

Comment 1. Line 10: The abbreviation "IPD" doesn't clearly reflect the text behind it. Could you please clarify?

Response 1: Thank you for pointing this out. IPD is meant to represent Invasive Streptococcus pneumoniae disease, or Invasive pneumococcal disease. We have added the word "disease," which hopefully better clarifies our intention of IPD.

Comment 2. Line 15: Add the year periods.

Response 2: This is a great point. We have adjusted the manuscript's abstract to include the dates.

Comment 3. In the conclusion, you mentioned "interventions implemented to reduce the spread of COVID-19." Please provide some examples.

Response 3:  Thank you for this comment, we agree that additional descriptions are important and have added two examples of those interventions. We were brief in our initial submission as the abstract submission was limited to 200 words.

Comment 4. Line 56: Spell out "MHS"

Response 4: Thank you for catching that we had not yet described the acronym yet.  This has been corrected.

Comment 5. Methods: Comparing IPD and IHI incidence rates to those if GBS wasn't described in the abstract. It seems like this is an essential part of your study since GBS infections are not prevented by vaccines.

Response 5: This is a great point, and while GBS is an important component of the study, we are limited to the 200 word limit in the abstract, and as such, it was left out to accommodate what we felt was the most impactful information about our study.

Comment 6. Line 71: How did you define the first two years? From March 2018 to February 2020? Please clarify in between parentheses.

Response 6: Definitions are absolutely important when discussing broad time periods. Yes, the first two years (pre-pandemic) were March 2018 to Feb 2020.  We have added and clarified that in the manuscript's methods section.

Comment 7: Line 76: Citation of results tables should be limited to the results section. Please remove it from this line.

Response 7: Thank you for this prudent point. It has been removed.

Comment 8. Lines 83-85: Please provide a reference for this sentence.

Response 8: We appreciate the reviewer catching this hanging statement without a reference. Our general goal was to use the south as a more temperate climate as the control; however, short of tropical and maybe sub-tropical regions of the world that have less seasonal variation, we did not find a strong enough reference to add. In order to avoid misleading our audience, we opted to remove this statement all together.

Comment 9. Line 86: I suggest making Table 3 as table 1 and moving it to follow this paragraph.

Response 9:  This is a great comment that will help the flow of the article for our future readers. This change was made and Table 3 is now Table 1 and has been moved up in the manuscript accordingly.

Comment 10. Lines 89-92: Separate this part into a standalone paragraph.

Response 10:  Thanks for this insightful comment to help parse out important information that otherwise gets buried in long paragraphs.  We agree and have made it a standalone paragraph.

Comment 11. Methods: it's better to divide this section into subsections with subheadings.

Response 11:  We agree that Methods and Materials section often are easier to follow. We added some short subsections, but certainly if the reviewer or editor felt that these could be even further improved - we are happy to do so.

Comment 12. Line 97: rate of care or incidence rate?

Response 12:  This was a typo and thank you for catching it.  We have simplified it to "IPD rates" (which now falls on line 123)

Comment 13. Table 1: Why is the full study population different between table 1a and table 1b?

Response 13:  This is a great question and certainly is one of the limitations of our repeated monthly cross sectional design and the use of our military data repository.  Since data was pulled monthly and the population of enrollees in the MHS changed each month, we had to simplify the full study population to age at which the children became enrolled in the MHS, and then when an ICD-10 was used, their age was updated for the cases.  For GBS, we didn't look beyond the newborn period, so the full study population, would have only included children enrolled right around birth. The GBS population closely aligns with the 0-5 month old age bracket in the IHI/IPD full study population table.

Comment 14. There's no description in the results for Figure 2.

Response 14: This is very true and quite an oversight on our end, as we are quite fond of our forest plots. We have gone back and added a few comments in each of the results subsections pointing the reader to Figure 2.

Comment 15. Line 210 and 224: Spell out "RSV" and "PCV"

Response 15:  Thanks for catching this oversight as we had not previously defined those acronyms. We have spelled those out in the revised manuscript.

Comment 16. Lines 231-234: This sentence describing H. influenzae should be moved to the introduction.

Response 16:  This is a good point by the reviewer.  After careful consideration, we opted to remove that sentence about H. flu as neither S. pneumoniae, nor GBS had the same level of granular microbiological detail. In retrospect, it did not necessarily add enough to our manuscript and therefore we felt comfortable without it.

Comment 17. Conclusion: Add a better emphasis on the importance of childhood vaccines at the end of the second paragraph.

Response 17: We completely agree with this comment as the importance of childhood vaccinations cannot be overstated.  From lines 355-357, we have added as the final line of our conclusion, "Given this resurgence in disease, appropriate and timely childhood immunizations are paramount to protecting this vulnerable population." We hope that the simplicity and straight forward sentence serves to reinforce how vital timely immunizations are in preventing invasive disease.

Round 2

Reviewer 2 Report

Comments and Suggestions for Authors

I appreciate the effort made by the authors to address all the comments with reflected edits in the manuscript. I have no further comments.